# Rapid Screening of Lipase Inhibitors in *Scutellaria*
*baicalensis* by Using Porcine Pancreatic Lipase Immobilized on Magnetic Core–Shell Metal–Organic Frameworks

**DOI:** 10.3390/molecules27113475

**Published:** 2022-05-27

**Authors:** Jinfang Xu, Pengkun Cao, Zhiyu Fan, Xujing Luo, Gangqiang Yang, Tingli Qu, Jianping Gao

**Affiliations:** School of Pharmacy, Shanxi Medical University, 56 Xinjian Road, Taiyuan 030001, China; xujinfang@sxmu.edu.cn (J.X.); caopengkun9257@163.com (P.C.); 13834405334@163.com (Z.F.); luoxujing08@163.com (X.L.); yanggangqiang2021@163.com (G.Y.); qutingli@126.com (T.Q.)

**Keywords:** ligand fishing, magnetic MOF, *Scutellaria baicalensis*, pancreatic lipase inhibitor

## Abstract

As for ligand fishing, the current immobilization approaches have some potential drawbacks such as the small protein loading capacity and difficult recycle process. The core–shell metal–organic frameworks composite (Fe_3_O_4_-COOH@UiO-66-NH_2_), which exhibited both magnetic characteristics and large specific surface area, was herein fabricated and used as magnetic support for the covalent immobilization of porcine pancreatic lipase (PPL). The resultant composite Fe_3_O_4_-COOH@UiO-66-NH_2_@PPL manifested a high loading capacity (247.8 mg/g) and relative activity recovery (101.5%). In addition, PPL exhibited enhanced tolerance to temperature and pH after immobilization. Then, the composite Fe_3_O_4_-COOH@UiO-66-NH_2_@PPL was incubated with the extract of *Scutellaria baicalensis* to fish out the ligands. Eight lipase inhibitors were obtained and identified by UPLC-Q-TOF-MS/MS. The feasibility of the method was further confirmed through an in vitro inhibitory assay and molecular docking. The proposed ligand fishing technique based on Fe_3_O_4_-COOH@UiO-66-NH_2_@PPL provided a feasible, selective, and effective platform for discovering enzyme inhibitors from natural products.

## 1. Introduction

Obesity, which is broadly defined as excess body weight for a given height, has become one of the major health concerns in the world, as it is an important risk factor for chronic metabolic diseases such as type 2 diabetes, hypertension, and dyslipidemia [1]. Pancreatic lipase is considered as a key enzyme in regulating the metabolism of lipids. It hydrolyses dietary lipids into small molecules of glycerol and fatty acids that the body can absorb [2]. Pancreatic lipase inhibitors, which can make pancreatic lipase lose part of the decomposition ability and thereby reduce the accumulation of fatty acids in the body, are considered as one class of effective anti-obesity agents [3]. Orlistat is the only lipase inhibitor diet drug currently in clinical use. Though it prevents about 30% of dietary fat from being absorbed by the body, the serious side-effects such as steatorrhea, fecal incontinence, and flatulence cannot be ignored [4]. Therefore, new alternative drug leads with few side-effects are needed.

Natural products compose a large chemical library with rich diversities in structures and bioactivities, from which a variety of compounds have been regarded as inhibitors for pancreatic lipase [5]. The roots of *Scutellaria baicalensis* Georgi, known as *Huang Qin* in Chinese, has been used in clinical treatment for more than 2000 years. *S. baicalensis* exhibits broad pharmacological activities including anti-inflammatory, antitumor, and hepatoprotective effects [6]. Its anti-obesity activity has also been reported, but the bioactive ingredients need further clarification [7]. Screening for active natural compounds is still very difficult as they exist in complex chemical systems and usually at very low concentrations. Traditional methods such as systematic separation and bioactivity-guided separation are usually labor-intensive, time-consuming, and inefficient [8].

Enzyme immobilization refers to the numerous techniques aiming to attach enzymes on solid matrices [9]. In comparison with native enzymes, the immobilized one exhibits several advantages including easy recovery and high stability [10,11]. The immobilized lipases have played an important role in biosensors, biodiesel production, and treatment of wastewater [12,13,14]. Ligand fishing, a new method for screening active ingredients from complex chemical systems, has been developed by using immobilized enzymes [15,16]. Until now, various types of solid supports, such as gold nanoparticles [17], magnetic nanoparticles [18], silica nanospheres [19], mesoporous fibers [20], and halloysite nanotubes [21], have been successfully used for the immobilization of target proteins through chemical or physical methods. However, current immobilization materials have some potential drawbacks such as small loading capacity and weak catalytic activity, which impedes the application of ligand fishing. Metal–organic frameworks (MOFs) are porous crystalline hybrid materials made by connecting inorganic and organic units by strong bonds, known for their large specific surface area, tunable pore size, and good biocompatibility [22,23]. Considering the advantage of MOFs, several studies have been recently devoted to enzyme immobilization on MOFs as the biological matrices for ligand fishing to obtain bioactive compounds from complex mixtures [24,25]. Nevertheless, after ligand fishing, the centrifugal separation of the MOFs immobilized enzyme is inconvenient and the regeneration process is difficult. The introduction of magnetism can overcome the restriction in materials recycling [22]. The lipase immobilized on magnetic MOFs (MMOFs) displayed a strong magnetic response and could be facilely separated by an external magnetic field. However, to the best of our knowledge, the reports about the application of MMOFs-enzyme composites for ligand fishing to obtain bioactive compounds from complex mixtures are very few.

Herein, we aimed to quickly screen lipase inhibitors from *S. baicalensis* by using porcine pancreatic lipase (PPL) immobilized on MMOFs. First, the core–shell MMOFs composite (Fe_3_O_4_-COOH@UiO-66-NH_2_) was fabricated, and PPL was encapsulated in a crystal lattice of UiO-66-NH_2_ to obtain Fe_3_O_4_-COOH@UiO-66-NH_2_@PPL. The MMOFs composite and immobilized lipase were characterized by multiple techniques. Specific parameters needed for immobilization were optimized to achieve high immobilization efficiency. The factors affecting the stability of the immobilized enzyme were compared. Moreover, we used a model mixture including lipase inhibitors and noninhibitors to verify the specificity of the developed method. Next, ligand fishing of lipase inhibitors based on Fe_3_O_4_-COOH@UiO-66-NH_2_@PPL from *S. baicalensis* was carried out. The compounds fished out were identified by HPLC-Q-TOF-MS/MS. Furthermore, the feasibility of this method was validated by an in vitro inhibitory activity assay as well as molecular docking. The synthesis route of the immobilized PPL and ligand fishing process was presented in Figure 1.

## 2. Results and Discussion

### 2.1. Characterization of the UiO-66-NH_2_ and Fe_3_O_4_-COOH@UiO-66-NH_2_@PPL

#### 2.1.1. Fourier Transform Infrared Analysis

The Fourier transform infrared (FT-IR) spectra of Fe_3_O_4_-COOH, Fe_3_O_4_-COOH@UiO-66-NH_2_, PPL, and Fe_3_O_4_@UiO-66-NH_2_@PPL are depicted in Figure 2. As for Fe_3_O_4_-COOH, the strong peak at 560 cm^−1^ was the characteristic absorption of Fe_3_O_4_ and could be assigned to the stretching vibration of Fe-O bonds [26]. In addition, a broad absorption band at 1560 cm^−1^ and the peak at 1410 cm^−1^ could be attributed to the C=O bond and -O-C-O- bond stretching vibration, respectively, confirming the modification of carboxyl onto Fe_3_O_4_ [27]. Additionally, peaks at 3456, 3353, and 1651 cm^−1^ were observed in the spectrum of Fe_3_O_4_-COOH@UiO-66-NH_2_, which, respectively, corresponded to the asymmetric and symmetric stretching modes, and bending vibration of N-H bonds [28], and the peaks at 1382 and 1254 cm^−1^ could be attributed to the stretching vibration and bending vibration of C-N bonds [29]. Meanwhile, the stretching vibration peak of Fe-O bonds (560 cm^−1^) could also be observed in the spectrum of Fe_3_O_4_-COOH@UiO-66-NH_2_, confirming the presence of magnetic Fe_3_O_4_. The N-H bending vibration peak at 1651 cm^−1^ of Fe_3_O_4_-COOH@UiO-66-NH_2_ shifted to 1620 cm^−1^ after immobilization, suggesting the formation of amide groups when PPL was bonded onto the surface of MOFs by cross-linking [30].

#### 2.1.2. X-ray Diffraction Analysis

The crystalline structure of Fe_3_O_4_-COOH, UiO-66-NH_2_, and Fe_3_O_4_-COOH@UiO-66-NH_2_ was determined by the powder X-ray diffraction (XRD) method. As displayed in Figure 3, the Fe_3_O_4_-COOH showed six characteristic X-ray diffraction peaks at 2θ of 30.1°, 35.5°, 43.1°, 53.4°, 57.0°, and 63°, respectively, corresponding to the crystal planes of Fe_3_O_4_ (220), (311), (400), (422), (511), and (440), and the diffraction pattern was found to be consistent with the pervious data [31]. The XRD pattern of the UiO-66-NH_2_ showed that the MOF was successfully synthesized with the high crystallinity, exhibiting sharp and intense characteristic peaks at 2θ of 7.4°, 8.6°, and 25.8°, which is the same as the literature [32]. The XRD pattern of the synthesized Fe_3_O_4_-COOH@UiO-66-NH_2_ matched well with that of the UiO-66-NH_2_ crystal and Fe_3_O_4_-COOH nanoparticles, confirming the excellent incorporation of Fe_3_O_4_-COOH into the UiO-66-NH_2_ lattice. It should be noted that the peaks related to UiO-66-NH_2_ were broadened as the particle size of MOF synthesized in the presence of Fe_3_O_4_-COOH was smaller than that of pure MOF (carboxylic acid groups acted as regulators, possibly controlling the nucleation and growth rates of UiO-66-NH_2_). In addition, the intensity of different peaks related to Fe_3_O_4_-COOH was decreased probably due to the formation of the core–shell structure.

#### 2.1.3. Transmission Electron Microscope Characterizations

The structures and morphologies of Fe_3_O_4_-COOH, UiO-66-NH_2_, and Fe_3_O_4_-COOH@UiO-66-NH_2_ were further illustrated by transmission electron microscope (TEM) characterizations, as shown in Figure 4. It could be seen that Fe_3_O_4_-COOH nanoparticles exhibited a spherical structure with a smooth surface and an average size of 150 nm, and were wrapped by a thin cover, which was a polyacrylate shell formed by the addition of sodium acrylate (Figure 4A). The pure UiO-66-NH_2_ nanoparticles had a regular octahedral structure with a particle size around 50~100 nm (Figure 4B), consistent with the morphology reported in previous literature [33]. The TEM image of Fe_3_O_4_-COOH@UiO-66-NH_2_ showed two distinct phases, that is, an irregularly shaped coating phase (MOF) and a core phase (Fe_3_O_4_-COOH), confirming the growth of UiO-66-NH_2_ crystals starting from the surface of Fe_3_O_4_-COOH to form the core–shell structure (Figure 4C,D). Meanwhile, the irregular shape may provide a larger surface area for adsorption as compared to the spherical shape.

### 2.2. Study on Optimum Immobilization Conditions

As the immobilization conditions had obvious effects on the enzyme catalytic activity and protein loading capacity, the parameters such as enzyme amount, concentration of cross-linking agents, and cross-linking time were studied to obtain the optimum enzyme-immobilized efficiency. As shown in Figure 5A, the effect of enzyme amount on immobilization efficiency was studied by changing the ratio of Fe_3_O_4_-COOH@UiO-66-NH_2_ to PPL (*w*/*w*). Five mass ratios of MMOF and PPL from 1:1 to 10:1 were investigated. The results manifested that the activity of the immobilized enzyme showed an upward trend for the ratio range of 1:1 to 2:1. When the ratio was higher than 2:1, instead, the immobilized enzyme activity decreased. The reason might be that the excess enzyme protein aggregated on the surface of MMOFs to cause accumulation and agglomeration, which reduced the space gap between enzyme molecules and hindered the interaction between the active center structure of enzyme and substrate molecules [34]. Meanwhile, the values of protein loading capacity decreased with the quantity of the MMOF, which was attributed to the protein loading of the immobilized enzyme reaching saturation due to a definite quantity of cross-linker and excess quantity of enzyme. Additionally, MMOF@PPL was prepared by varying the concentration of cross-linking agent. Glutaraldehyde (GA) was herein selected as the cross-linking agent because of its small molecular weight. GA could easily enter the gap between the enzyme molecules and combine with the amino group to form more “Schiff base”, which increased the mechanical stability of the enzyme [35]. As shown in Figure 5B, with the increasing concentration of GA (110~170 mM), the protein loading of the immobilized enzyme was slightly increased and then stabilized at about 240 mg/g. However, the activity of the immobilized enzyme first showed an upward trend with the increase in GA concentration, reaching the highest activity value at the concentration of 150 mM, and then began to decrease. The reason might be that excessive GA could also cross-link enzyme proteins, destroy the secondary structure of enzyme proteins, and denaturant them [36]. Figure 5C shows the effect of different cross-linking times on the immobilized enzyme. With the extension of the cross-linking reaction time (0.5~4 h), the protein loading of immobilized PPL showed a gradual upward trend and tended to be saturated when the cross-linking time exceeded 1 h, but the immobilized enzyme activity reached a maximum at 2 h due to the greater enzymatic protein toxicity caused by GA with longer cross-linking time. Overall, according to the above analysis results, the optimal lipase immobilization conditions were as follows: MMOFs-to-enzyme mass ratio of 2:1, GA concentration of 150 mM, and cross-linking time of 2 h. The protein loading capacity achieved 247.8 mg/g, and the recovery activity of the immobilized enzyme was 101.5% under the above-mentioned conditions. Compared with UiO-66-NH_2_ and Fe_3_O_4_-COOH nanoparticles, the composite material Fe_3_O_4_-COOH@UiO-66-NH_2_ had a larger specific surface area due to the existence of numerous convex and concave pores on the surface, so it provided more binding sites for the enzyme and resulted in larger protein loading. Moreover, the higher immobilized enzyme activity could result from the better stability of the PPL immobilized on the surface of MMOFs [37].

### 2.3. Stability of Immobilized PPL

Temperature and pH value were critical factors affecting the activities of the immobilized lipase. As shown in Figure 6A, the enzymatic activity of free and immobilized lipase was determined at different temperatures from 20 °C to 65 °C in PBS of pH 7.5. The immobilized pancreatic lipase reached the maximum activity at 45 °C, which was higher than that of the free enzyme (37 °C). In addition, the decreasing rate of relative activity for free lipase was much faster than that of the immobilized lipase with the increase in temperature ranging from 45 °C to 65 °C, which indicated that the immobilized lipase displayed a better thermal stability and higher thermal resistance as compared to the free one. As shown in Figure 6B, the immobilization pH value varying from 6.0 to 9.0 was investigated at 37 °C. The activities of the immobilized and free enzyme increased with the pH from 6.0 to 7.5, and both reached their maximum at pH 7.5. However, the relative activity of immobilized lipase was greater in the acid or alkali region compared with the free one, indicating the higher pH stability of Fe_3_O_4_@UiO-66-NH_2_@PPL. These above results illustrated that the immobilization of lipase increased the mechanical strength and decreased the configuration variability [38].

### 2.4. Verification of Ligand Fishing

In order to verify the specificity and feasibility of the synthetic immobilized PPL, a standard mixture model consisting of hesperidin (lipase inhibitor, IC_50_ = 64.92 µM, as positive control) and gallic acid (no lipase inhibitory activity, as negative control) was designed for validation [24,39]. The HPLC analysis results of each eluent are shown in Figure 7, where S1 is the chromatogram of the model mixture, and S2 is the chromatogram of the methanol elution of the MMOF after ligand fishing. It could be seen from S2 that only hesperidin was screened out by the immobilized enzyme, while gallic acid failed to bind specifically to the immobilized enzyme. The results indicated that it was feasible to use Fe_3_O_4_-COOH@UiO-66-NH_2_@PPL to screen active small molecules from traditional Chinese medicine extracts.

### 2.5. Ligand Fishing of PPL Inhibitors from S. baicalensis 

This established approach was applied to fish out potential PPL inhibitors from *S. baicalensis*. As shown in the UPLC-Q-TOF-MS/MS chromatogram of the extract of *S. baicalensis*, 27 compounds were detected (Figure 8, line a). In comparison, 8 compounds as potential inhibitors of PPL were detected after ligand fishing with Fe_3_O_4_-COOH@UiO-66-NH_2_@PPL (Figure 8, line b). According to the mass spectrometry data, compounds 1~8 were identified as 2′, 3, 5, 6′, 7-pentahydroxyflavonoids, scutellarein, baicalin, oroxyloside, wogonoside, skullcapflavone Ⅱ, wogonin, and oroxylin A, respectively (Table 1 and Appendix A) [40,41,42]. In addition, in order to verify whether compounds bind to the immobilized enzyme by other means such as physical adsorption, resulting in false positive inhibition, a blank control was set-up in this experiment. The results showed that only wogonin (compound 7) had a certain nonspecific adsorption with the blank MMOF (Figure 8, line c).

### 2.6. In Vitro Enzyme Inhibition Activity Analysis and Molecular Docking Study

In order to verify the validity of the method based on PPL@MMOF ligand fishing, three of the screened compounds, including baicalin, wogonin, and oroxylin A, were chosen to test the PPL inhibitory activity by the *p*-nitrophenol method with orlistat as the positive control. The results showed that the IC_50_ values of orlistat, baicalin, wogonin, and oroxylin A were 1.9 ± 0.6 µM, 242.2 ± 5.6 µM, 168.5 ± 8.4 µM, and 82.7 ± 3.6 µM, respectively. Though the IC_50_ values were much higher than that of the positive drug orlistat, these ligands, especially oroxylin A, could be considered as effective natural lipase inhibitors. The results also proved the superiority of the immobilized lipase on MMOF with higher protein loading and enzymatic activity for ligand fishing.

Molecular docking was carried out to better explain the binding mechanism and modes between eight ligands and PPL. As shown in Table 2, the CScore values of ligands 1 to 8 were all greater than four, indicating their potential PPL inhibitory activities. Among them, wogonoside had the highest CScore value, and its 3D and 2D docking modes are illustrated in Figure 9. It was clearly shown that wogonoside had been inserted into the active site of PPL and formed three hydrogen bonds through the interaction of phenolic hydroxyl and glycosyl with PHE77, PHE215, and SER152 of PPL. The interactions of ligands with the amino acid residues of PPL displayed by molecular docking studies revealed the possible inhibition mechanism of these bioactive compounds.

## 3. Materials and Methods

### 3.1. Materials and Reagents

Ferric chloride hexahydrate (FeCl_3_·6H_2_O; AR, 99%), sodium acrylate (CH_2_=CHCOONa; 95%), ZrCl_4_ (99.5%), aminoterephthalic acid (ATA; 98%), N, N-dimethyl-formamide (DMF; AR, 99.5%), ethylene glycol (EG; AR, 98%), anhydrous sodium acetate (CH_3_COONa; AR, 99%), coomassie brilliant blue G-250 (AR, 98%), *p*-nitrophenol (AR), and *p*-nitrophenyl butyrate (*p*-NPB; 98%) were purchased from Shanghai Macklin Biochemical Co., Ltd., (Shanghai, China). In addition, 50% glutaraldehyde (GA), acetic acid (HAc), ethanol absolute, and ammonium sulfate were of analytical grade and purchased from Shanghai Aladdin Biochemical Technology Co., Ltd., (Shanghai, China). Pancreatic lipase (EC.3.1.1.3 from porcine pancreas) and bovine albumin (BSA; 98%) were provided by Shanghai Yuanye Biotechnology Co., Ltd., (Shanghai, China). *S*. *baicalensis* extract was purchased from Beijing Kangrentang Pharmaceutical Co., Ltd., (Beijing, China). All chemicals and reagents were obtained from commercial sources and used without any further purification.

### 3.2. Experimental Instrument

The HPLC analysis was carried out using a Shimadzu LC-2030 series HPLC instrument with a flow rate of 1.0 mL/min, column temperature of 40 °C, and UV detection wavelength of 273 nm. The samples were subjected to UPLC-Q-TOF-MS/MS analysis on an ACQUITY UPLC H-Class instrument coupled with a Thermo Q Exactive Mass Spectrometer via an API source. FT-IR spectra were obtained on a Nicolet iS5 Fourier transform infrared spectrometer device with the analytical wavenumber range of 4000~400 cm^−1^. Powder XRD patterns were recorded by an D8 Advance A25 X-ray diffractometer with Ni-filtered Cu Kα radiation and 2θ in the range between 5° and 90°. The sample morphologies were analyzed utilizing a JEM-2100F high-resolution field emission TEM operating at an accelerating voltage of 200 kV. Enzyme activity analysis was performed by using a SpectraMax M5 multi-function microplate reader at a wavelength of 405 nm.

### 3.3. Preparation of Fe_3_O_4_-COOH@UiO-66-NH_2_@PPL

#### 3.3.1. Synthesis of Fe_3_O_4_-COOH Nanoparticles

The carboxylic-modified Fe_3_O_4_ nanoparticles (Fe_3_O_4_-COOH) were in situ-synthesized as previously reported with the solvothermal method with some modifications [43]. FeCl_3_·6H_2_O (1.08 g) was dispersed in a beaker containing 40 mL of ethylene glycol and ultrasonically shaken until it was completely dissolved. Sodium acrylate (1.6 g) and NaAc (0.8 g) were added to the solution in sequence and ultrasonicated for 20 min. Then, the mixture solution was stirred at 50 °C for 1 h to obtain a homogeneous mixture solution. Subsequently, the resulting mixture was transferred to a 100 mL stainless-steel Teflon-lined autoclave to heat at 200 °C for 10 h. After cooling to room temperature, the black magnetic micro-spheres were washed with deionized water and ethanol, dried at 60 °C, and denoted as Fe_3_O_4_-COOH.

#### 3.3.2. Synthesis of Fe_3_O_4_-COOH@UiO-66-NH_2_ and UiO-66-NH_2_ Nanoparticles

The magnetic Fe_3_O_4_-COOH@UiO-66-NH_2_ nanoparticles were prepared through a facile solvothermal method [44]. Fe_3_O_4_-COOH (0.2 g) prepared above was dispersed into 20 mL of DMF and sonicated for 30 min. Meanwhile, ZrCl_4_ (0.1 mol), ATA (0.1 mol), and HAc (2 mL) were dispersed in 30 mL of DMF by ultrasound for 30 min to form a MOF precursor solution, which was then added to the Fe_3_O_4_-COOH solution and sonicated for another 30 minutes. Subsequently, the resulting mixture was transferred into a 100 mL stainless-steel Teflon-lined autoclave and heated at 120 °C for 24 h. After the reaction, the magnetic nanomaterials were alternately washed with DMF and water for 3 times, and then dried in a vacuum at 80 °C for 6 h. The resulting yellow-brown powder was the MMOF Fe_3_O_4_-COOH@UiO-66-NH_2_. The neat UiO-66-NH_2_ nanoparticles were prepared according to the above process without using Fe_3_O_4_-COOH.

#### 3.3.3. Immobilizing PPL on Fe_3_O_4_-COOH@UiO-66-NH_2_

The immobilized lipase was synthesized by the precipitation cross-linking method. PPL immobilized onto Fe_3_O_4_-COOH@UiO-66-NH_2_ was prepared by the precipitation cross-linking method as the reported method with some modifications [30]. Briefly, the as-prepared Fe_3_O_4_@UiO-66-NH_2_ (10 mg) was mixed with 1 mL of crude PPL solution (100 mg of crude protein in 1 mL of 50 mM PBS containing 5.11 mg of PPL, pH 7.5). Then, a 4 mL saturated ammonium sulfate solution was added in the mixture to precipitate the lipase under mechanical stirring for 30 min at 4 °C. Subsequently, 158 µL of 50% glutaraldehyde (GA) was injected dropwise into the above solution, followed by cross-linking reaction for 3 h with agitation. Finally, the magnetic nanoparticles in the resulting suspension were collected by an external magnet, washed 3 times with PBS (50 mmol/L, pH 7.5) and then stored at 4 °C for subsequent use.

### 3.4. Optimization of Immobilization Conditions

The amount of immobilized lipase on Fe_3_O_4_-COOH@UiO-66-NH_2_ was determined by calculating the difference in protein content in the supernatant before and after adsorption by the Bradford method [45]. The formula for calculating the protein loading capacity (mg/g) was as follows: Amount of protein load = M_e_ (mg)/M_MMOF_ (g), where the M_e_ represents the amount of enzyme protein loaded on the carrier material after reaction, and the M_MMOF_ represents the amount of added carrier material [46]. The catalytic activity of the immobilized lipase was assessed by comparing free lipase with the hydrolysis of *p*-NPB using the previously reported *p*-nitrophenol method [47]; the general process is as follows: the quantitative enzyme protein was weighed and added with 50 mM PBS buffer, 50 µL of 100 mM *p*-NPB substrate solution was added to 1 mL and 2 mL of absolute ethanol after the reaction at 37 °C for 30 min to terminate the reaction, and the absorbance at a wavelength of 405 nm was recorded. Meanwhile, the amount of enzyme required to catalyze the release of 1 mM *p*-nitrophenol per minute at 37 °C was defined as one unit of enzyme activity (U). In addition, the relative activity (%) was defined by the following equation: Relative activity (%) = (B/A) * 100, where A are the samples that reacted under the optimized conditions, and B are the samples that reacted under the harsh conditions [48].

#### 3.4.1. The Mass Ratio of MMOF to PPL

Different amounts of MMOF and enzyme protein were weighed to make the mass ratios of 1:1, 2:1, 4:1, 5:1, and 10:1 and then the immobilized enzymes were synthesized in the cross-linking reaction for 2 h under the condition of the cross-linking agent concentration of 150 mM. The protein loading and enzyme activity of the immobilized enzymes synthesized under different mass ratio conditions were determined according to the above-mentioned Bradford method and *p*-nitrobenzene method.

#### 3.4.2. Concentration of Crosslinking Agent

The quantitative MOF and enzyme protein were weighed to make the mass ratio of 2:1, and then the immobilized enzyme was synthesized by cross-linking reaction for 2 h under the conditions of cross-linking agent concentrations of 110, 130, 140, 150, and 170 mM. The protein loading and enzyme activity of the immobilized enzymes synthesized under different concentrations of crosslinking agent conditions were determined according to the above-mentioned Bradford method and *p*-nitrobenzene method.

#### 3.4.3. Cross-Linking Time

The quantitative MOF and enzyme protein were weighed to make the mass ratio of 2:1, and then the immobilized enzyme was synthesized by cross-linking reaction for 0.5, 1, 2, 3, and 4 h under the condition of cross-linking agent concentrations of 150 mM, respectively. The protein loading and enzyme activity of the immobilized enzymes synthesized under different cross-linking time conditions were determined according to the above-mentioned Bradford method and *p*-nitrobenzene method.

### 3.5. Study on the Stability of the Immobilized and Free Lipase

The effects of temperature and pH on the activities of free and immobilized lipase were studied. Each free and immobilized lipase solution was kept in PBS of pH 7.5 at temperatures ranging from 20 °C to 65 °C for 20 min. In addition, each free and immobilized lipase solution was prepared at 37 °C in PBS under a variety of pH values ranging from 6 to 9 for 20 min. Then, the enzyme activities were calculated by hydrolysis of *p*-NPB.

### 3.6. Verification of Ligand Fishing

The hesperidin and gallic acid were mixed in aqueous solution (dissolved by a trace amount of DMSO) with a concentration of 200 mg/L as the model mixture, denoted as S1. To 20 mg of Fe_3_O_4_-COOH@UiO-66-NH_2_@PPL, 1 mL of the S1 was added, and the resulting solution was incubated for 90 min at 37 °C. After a magnetic separation, the MMOFs were washed with 3 × 1 mL PBS (50 mM, pH = 7.5) solution to remove nonspecifically bound compounds. Subsequently, the 95% methanol solution (3 × 1 mL) was added to degenerate the PPL and release the compounds specifically bound by the immobilized enzyme, yielding eluent S2. The collected S1 and S2 solutions were chromatographically analyzed on a YMC-Pack C_18_ column (4.6 × 150 mm, 5 µm). The mobile phase consisted of solvent A (aqueous solution) and B (methanol); the gradient elution conditions was as follows: 0~40 min (20~100% B).

### 3.7. Application of Ligand Fishing in S. baicalensis

The PBS solution (3 mL) containing 50 mg of *S. baicalensis* extract was prepared to incubate with Fe_3_O_4_-COOH@UiO-66-NH_2_@PPL (20 mg) for 90 min at 37 °C. Then, the MMOFs were separated by magnetic adsorption and washed 3 times with 1 mL of PBS buffer to remove nonspecifically bound compounds. Subsequently, the incubated MMOFs were washed with 95% methanol (3 × 1 mL) to denature lipase and release the specifically bound compounds. The eluate was collected and analyzed by UPLC-Q-TOF-MS/MS with a Thermo Hypersil GOLD C_18_ column (100 × 2.1 mm, 1.9 µm). The mobile phase consisted of solvent A (formic acid water) and B (methanol). The gradient elution conditions were as follows: 0~1 min (20~30% B), 1~3 min (30~40% B), 3~7 min (40~50% B), 7~12 minutes (50~55% B), 12~17 minutes (55~70% B), 17~20 minutes (70~95% B), 20~22 minutes (95% B). Meanwhile, the same incubation for Fe_3_O_4_-COOH@UiO-66-NH_2_ unrelated PPL was performed as a negative control to distinguish the nonspecific binding compounds. Finally, the 95% methanol eluate was collected and analyzed under the same chromatographic conditions.

### 3.8. In Vitro Validation of Lipase Inhibitory Activity

Taking orlistat as a positive control, the method for the determination of lipase inhibitory activity was determined as previously described [49]. The PPL was dissolved in 50 mM PBS buffer (pH 7.5), yielding the final enzyme solution (1 mg/mL). The samples were dissolved in 1 mL of DMSO and then diluted with an appropriate amount of PBS buffer (pH 7.5) into different concentration gradients. Subsequently, 50 µL of inhibitors at different concentrations and 200 µL of lipase solution were dissolved in 700 µL of 50 mM PBS buffer (pH 7.5). After incubation at 37 °C for 10 min, 50 µL of the substrate *p*-NPP (10 mM) was then added to initiate the enzymatic reaction. After incubation at 37 °C for 30 min, the reaction was terminated by adding 2 mL of anhydrous ethanol, and the absorbance at a wavelength of 405 nm was recorded to calculate their IC_50_ values. Orlistat was used as a positive control. The experiments in each group were repeated three times.

### 3.9. Molecular Docking

The crystal structure of pancreatic lipase protein (PDB code: 1lpb) was obtained from the PDB database, and then the selected compounds were analyzed by SYBYL-X 2.1.1 software after removing free water and hydrogenation by SYBYL-X 2.1.1 software; meanwhile, the analysis results were plotted with PyMOL.

## 4. Conclusions

In this study, core–shell-structured Fe_3_O_4_-COOH@UiO-66-NH_2_ had been successfully synthesized and employed as a solid support for the immobilization of PPL. The prepared composite exhibited advantages of large specific surface area, an abundance of active sites, as well as the superparamagnetic property. Under the optimal conditions, the load amount of PPL was 247.8 mg/g, and the relative activity recovery was 101.5%. The stability of the immobilized enzyme was improved. In addition, a convenient strategy incorporating ligand fishing based on Fe_3_O_4_-COOH@UiO-66-NH_2_@PPL and UPLC-Q-TOF-MS/MS was established for the purpose of efficiently and reliably screening bioactive compounds from natural products. Based on the above method, eight inhibitors were screened and identified from the traditional Chinese medicine *S. baicalensis*. The feasibility of this method was proved by the PPL inhibitory activity assay and the molecular docking. Taken together, this strategy is expected to be applied to screen potential enzyme inhibitors from natural products, which will accelerate the discovery of the new drug candidates.

## Figures and Tables

**Figure 1 molecules-27-03475-f001:**
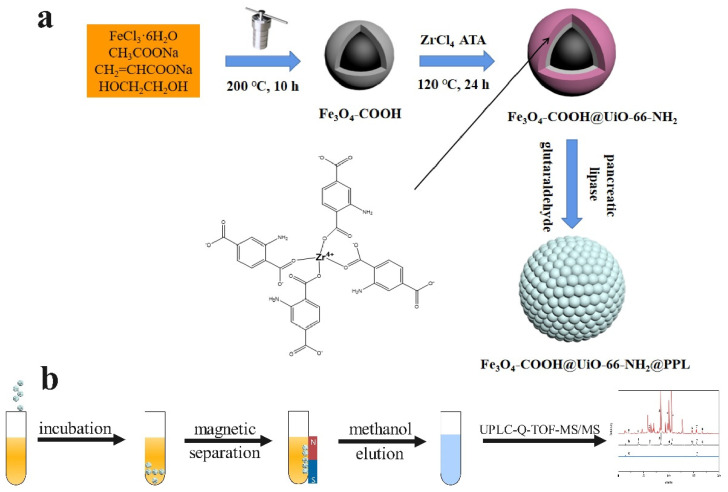
Schematic diagram of the synthesis of Fe_3_O_4_-COOH@UiO-66-NH_2_@PPL (**a**) and ligand fishing process (**b**).

**Figure 2 molecules-27-03475-f002:**
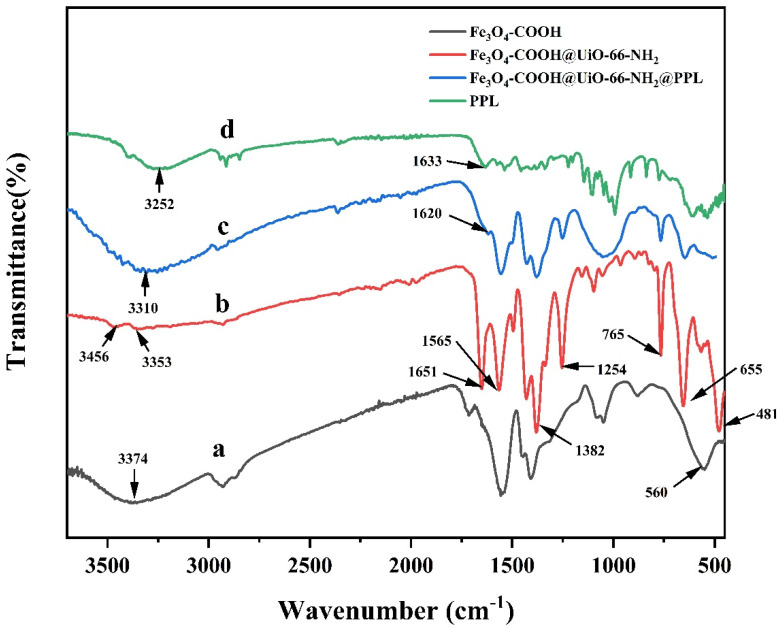
The FT-IR spectra of Fe_3_O_4_-COOH (**a**), Fe_3_O_4_-COOH@UiO-66-NH_2_ (**b**), Fe_3_O_4_-COOH@UiO-66-NH_2_@PPL (**c**), and PPL (**d**).

**Figure 3 molecules-27-03475-f003:**
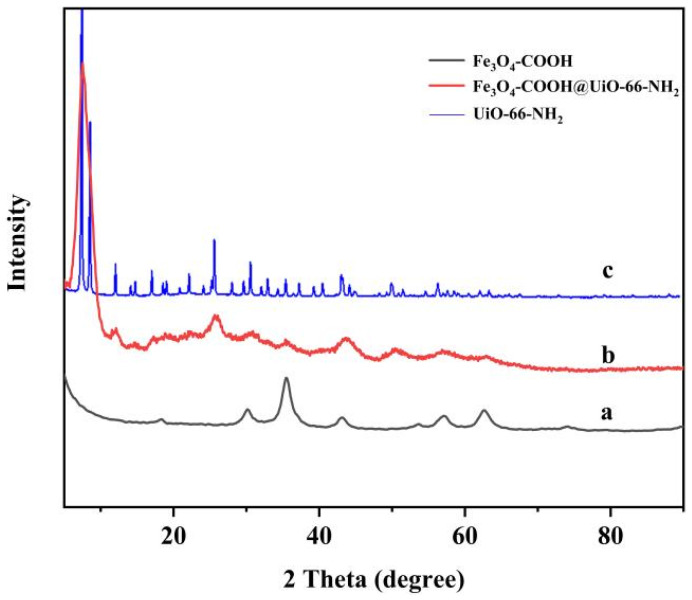
The XRD patterns of Fe_3_O_4_-COOH (**a**), Fe_3_O_4_-COOH@UiO-66-NH_2_ (**b**), and UiO-66-NH_2_ (**c**).

**Figure 4 molecules-27-03475-f004:**
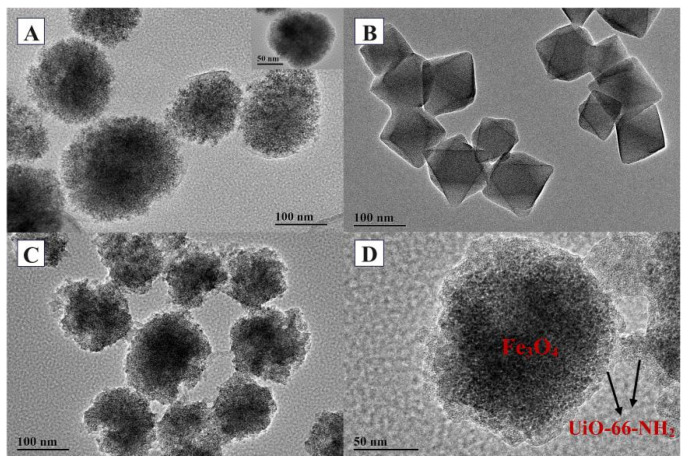
TEM images of Fe_3_O_4_-COOH (**A**), UiO-66-NH_2_ (**B**), Fe_3_O_4_-COOH@UiO-66-NH_2_ (**C**,**D**).

**Figure 5 molecules-27-03475-f005:**
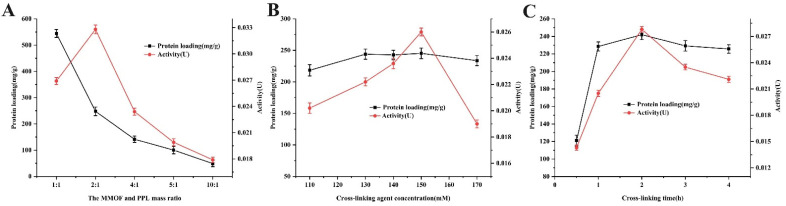
The effects of the MMOF and PPL mass ratio (**A**), glutaraldehyde concentration (**B**), and the cross-linking time (**C**) on protein loading and activity of immobilized PPL.

**Figure 6 molecules-27-03475-f006:**
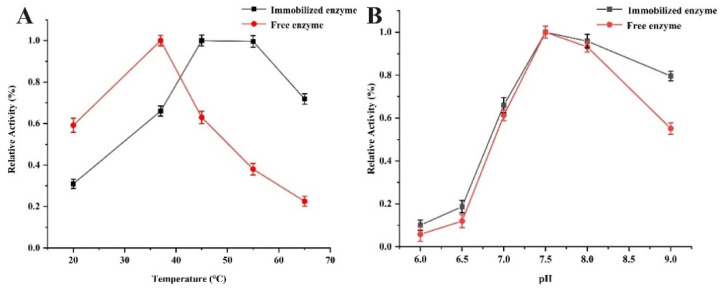
The effects of the temperature (**A**) and the pH (**B**) on the relative activity of free PPL and Fe_3_O_4_@UiO-66-NH_2_@PPL.

**Figure 7 molecules-27-03475-f007:**
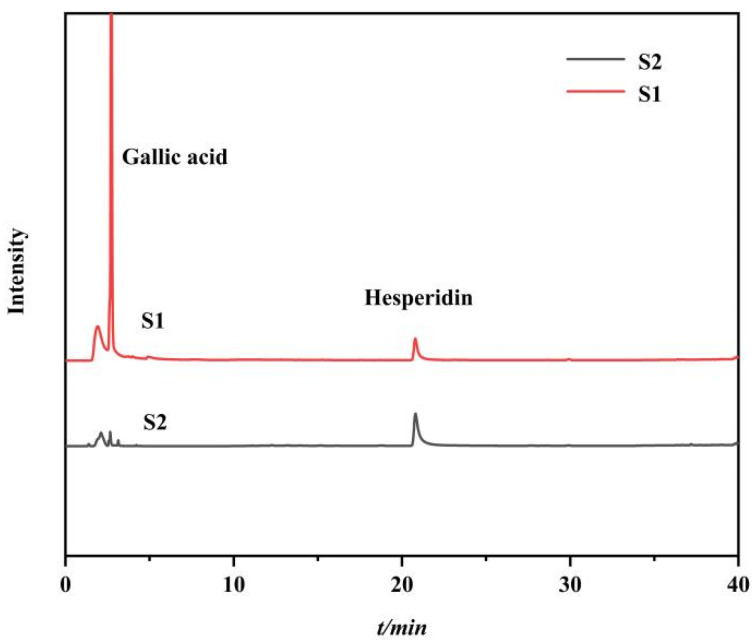
HPLC diagrams of model mixture (**S1**) and methanol eluent of the ligand fishing by Fe_3_O_4_-COOH@UiO-66-NH_2_@PPL (**S2**).

**Figure 8 molecules-27-03475-f008:**
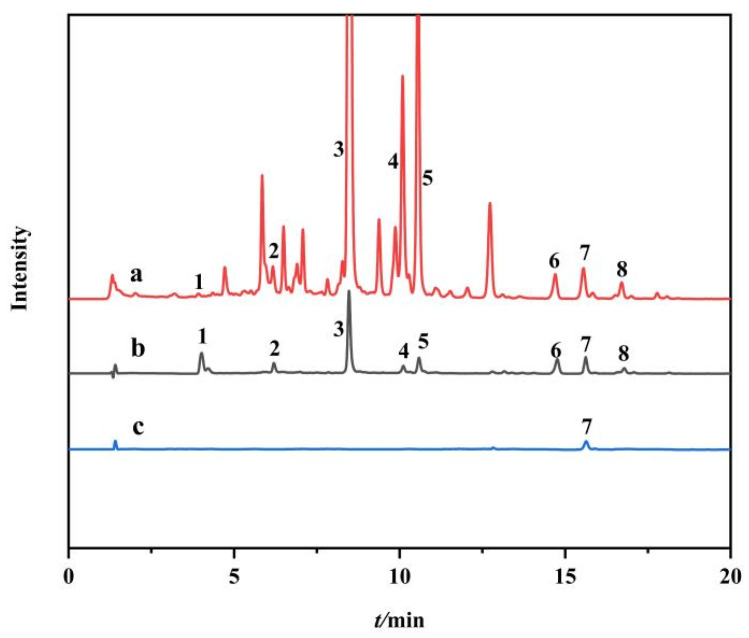
UPLC-Q-TOF-MS/MS diagrams of the water solution of *S. baicalensis* (**a**), the methanol eluent of the ligand fishing by Fe_3_O_4_-COOH@UiO-66-NH_2_@PPL (**b**), and methanol eluent of the ligand fishing by blank MMOFs (**c**).

**Figure 9 molecules-27-03475-f009:**
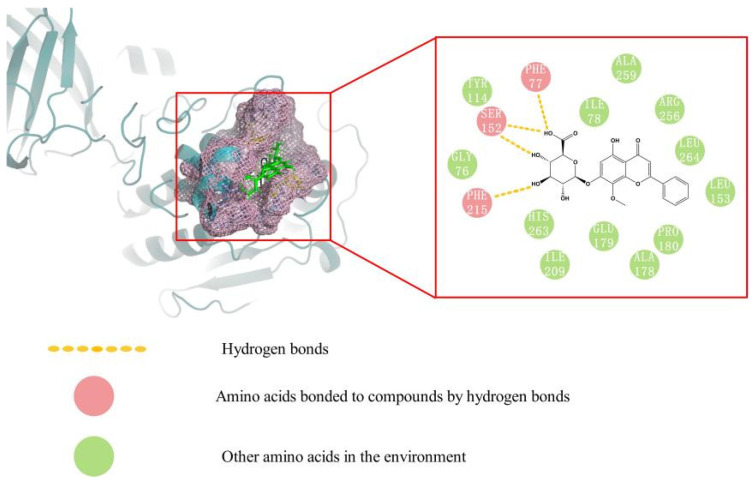
Molecular docking model diagram of wogonoside.

**Table 1 molecules-27-03475-t001:** MS/MS fragment ions of 8 compounds obtained by ligand fishing.

Peak	*t_R_*/min	m/z, [M+H] ^+^	MS/MS	Compounds
1	4.02	305.0658	285.0403, 275.0561, 259.0610, 177.0195, 125.0246	2′, 3, 5, 6′, 7- Pentahydroxyflavanone
2	6.20	287.0551	166.9987, 139.0039	Scutellarein
3	8.47	447.0924	271.0615	Baicalin
4	10.11	461.1079	285.0771, 270.0538	Oroxyloside
5	10.58	461.1081	285.0757, 270.0522	Wogonoside
6	14.76	375.1075	360.0846, 345.0605, 327.0500, 197.0082	Skullcapflavone Ⅱ
7	15.63	285.0758	270.0435	Wogonin
8	16.78	285.0758	270.0434	Oroxylin A

**Table 2 molecules-27-03475-t002:** The results of docking analysis of 8 compounds obtained by ligand fishing.

NO.	Compounds	CScore	Interaction Residues
1	2′, 3, 5, 6′, 7-Pentahydroxyflavanone	5.7782	ASP311, CYS39, ZYS239, ASP247
2	Scutellarein	4.5160	TYR114, PHE77, LEU153, SER152
3	Baicalin	5.9907	PHE77, PHE215, SER152
4	Oroxyloside	5.3949	ASP311, ARG265, ASP249, ASP247, ASP257
5	Wogonoside	6.2375	PHE77, PHE215, SER152
6	Skullcapflavone Ⅱ	5.4781	PHE77, SER152
7	Wogonin	5.3444	ARG38, ZYS239, GLY1, ASP331
8	Oroxylin A	4.5557	LEU153, SER152, PHE77

## Data Availability

Not applicable.

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
