# Peer review of "Rapid Screening of Lipase Inhibitors in Scutellaria baicalensis by Using Porcine Pancreatic Lipase Immobilized on Magnetic Core–Shell Metal–Organic Frameworks"

_molecules, 2022, doi:10.3390/molecules27113475_

Round 1

Reviewer 1 Report

Xu and co-authors reported the development of core-shell metal organic frameworks (MOFs) composite. Then, porcine pancreatic lipase (PPL) was encapsulated on synthetized MOFs.  The produced material was characterized by different techniques such as infrared, X-ray, and TEM. Moreover, the authors evaluated the activity of PPL encapsulated on MOFs and free PPL regarding pH and temperature. The authors also reported molecular docking studies.  Finally, the authors evaluated the composite Fe3O4-COOH@UiO-66-NH2@PPL incubated with extract of Scutellaria baicalensis to fish out the ligands. Eight lipase inhibitors were obtained and identified by UPLC-Q-TOF-MS/MS.

I suggest the publication of this manuscript after address the following issue:

  1. Please add the chromatograms and mass spectra analysed as supplementary material.

Author Response

Comment : Please add the chromatograms and mass spectra analyzed as supplementary material.

Response: Thank you for your precious suggestion. We have supplemented the chromatograms and mass spectra of the eight compounds in the supplementary material. The fragmentation behaviors of these compounds have also been analyzed.

Reviewer 2 Report

Dear authors,

The manuscript entitled "Rapid Screening of Lipase Inhibitors in Scutellaria Baicalensisby Using Porcine Pancreatic Lipase Immobilized on Magnetic Core-Shell Metal-Organic Frameworks" is an interesting work on novel methods of screening for the presence of lipase inhibitors. It is written in an accessible language and it also manages to be meritotically correct. However, I have a few questions and suggestions. 

The results:
My main concern is the lack of even basic statistical analysis of the results obtained. At least the standard deviations should be shown in the charts. (Fig 5-6). Analysis of variance is advisable.

Materials and methods:

Line 323 - 324 "Sub Consequently, 158 µL of 50% glutaraldehyde (GA) was injected dropwise into the above solution, followed by cross-linking reaction (...) " Has any compound been used to inactivate free aldechide reactive groups? This is advisable to avoid non-specific effects.

Minor editorial notes:
The text formatting in the manuscript should be standardized (line spacing).

Yours faithfully,

Author Response

Comment 1: My main concern is the lack of even basic statistical analysis of the results obtained. At least the standard deviations should be shown in the charts. (Fig 5-6). Analysis of variance is advisable.

Response: Much appreciated for your positive comment. As suggested, we have analyzed the standard deviation of the obtained data for each set of results, and clearly characterized it in the revised graphs by error bars.

Comment 2: Line 323 - 324 "Sub Consequently, 158 µL of 50% glutaraldehyde (GA) was injected dropwise into the above solution, followed by cross-linking reaction (...) " Has any compound been used to inactivate free aldechide reactive groups? This is advisable to avoid non-specific effects.

Response: Thanks. It has been reported that hydroxylamine hydrochloride could be utilized to react with free aldehyde groups [1]. As for the precipitation cross-linking method we used, almost all of the enzymes were deposited on the surface of MMOFs during the precipitation process. Therefore, GA could quickly connect the enzyme with MMOFs. Considering the small amount of GA and the huge amount of enzyme coated on the surface of MMOF, the effect of free aldehyde groups on enzyme activity and subsequent ligand fishing could be ignored [2-3]. That is the reason why we did not conduct inactivation treatment.

Reference:

  1. Wang, Z.; Li, X.Q.; Chen, M.H.; Liu, F.Y.; Han, C.; Kong, L.Y.; Luo, J.G. A strategy for screening of α-glucosidase inhibitors from Morus alba root bark based on the ligand fishing combined with high-performance liquid chromatography mass spectrometer and molecular docking. Talanta. 2018, 180, 337-345. https://doi.org/10.1016/j.talanta.2017.12.065
  2. Cao, S.L.; Yu, D.M.; Li, X.; Smith, T.J.; Li, N.; Zong, M.H.; Wu, H.; Ma, Y.Z.; Lou, W.Y. Novel nano-/micro-biocatalyst: soybean epoxide hydrolase immobilized on UiO-66-NH2 MOF for efficient biosynthesis of enantiopure (R)-1, 2-Octanediol in deep eutectic solvents. Acs Sustain. Eng. 2016, 4, 3586-3595. https://doi.org/10.1021/acssuschemeng.6b00777
  3. Cao, S.L.; Xu, H.; Li, X.H.; Luo, W.Y.; Zong, M.H. Papain@Magnetic Nanocrystalline Cellulose Nanobiocatalyst: A highly efficient biocatalyst for dipeptide biosynthesis in deep eutectic solvents. ACS Sustainable Chem. Eng. 2015, 3, 1589-1599. https://doi.org/10.1021/acssuschemeng.5b00290

Comment 3: The text formatting in the manuscript should be standardized (line spacing).

Response: Thank you for your comments. The format has been carefully examined and corrected in the revised manuscript.